

# HSRL-2 aerosol optical measurements and microphysical retrievals vs. airborne in situ measurements during DISCOVER-AQ 2013: an intercomparison study

Patricia Sawamura[1,2], Richard H. Moore[1], Sharon P. Burton[1], Eduard Chemyakin[1,2], Detlef Müller[3,2], Alexei Kolgotin[4], Richard A. Ferrare[1], Chris A. Hostetler[1], Luke D. Ziemba[1], Andreas J. Beyersdorf[5], and Bruce E. Anderson[1]

[1]NASA Langley Research Center, Hampton, VA, USA
[2]Science Systems and Applications, Inc., Hampton, VA, USA
[3]University of Hertfordshire, Hatfield, Hertfordshire, UK
[4]Physics Instrumentation Center, Troitsk, Russia
[5]California State University, San Bernardino, CA, USA

*Correspondence to:* P. Sawamura (patricia.sawamura@nasa.gov)

**Abstract.** We present a detailed evaluation of remotely-sensed aerosol microphysical properties obtained from an advanced, multi-wavelength High Spectral Resolution Lidar (HSRL-2) during the 2013 NASA DISCOVER-AQ field campaign. Vertically-resolved retrievals of fine mode aerosol number, surface area, and volume concentration as well as aerosol effective radius are compared to 108 co-located, airborne in situ measurement profiles in the wintertime San Joaquin Valley, California, and in summertime Houston, Texas. An algorithm for relating the dry in situ aerosol properties to those obtained by the HSRL at ambient relative humidity is discussed. We show that the HSRL-2 retrievals of ambient fine mode aerosol surface area and volume concentrations agree with the in situ measurements to within 25% and 10%, respectively, once hygroscopic growth adjustments have been applied to the dry in situ data. Despite this excellent agreement for the microphysical properties, extinction and backscatter coefficients at ambient relative humidity derived from the in situ aerosol measurements using Mie theory are consistently smaller than those measured by the HSRL, with average differences of 31% ± 5% and 53% ± 11% for California and Texas, respectively. This low bias in the in situ estimates is attributed to the presence of coarse mode aerosol that are detected by HSRL-2 but that are too large to be well sampled by the in situ instrumentation. Since the retrieval of aerosol volume is most relevant to current regulatory efforts targeting fine particle mass (PM$_{2.5}$), these findings highlight the advantages of an advanced $3\beta+2\alpha$ HSRL for constraining the vertical distribution of the aerosol volume or mass loading relevant for air quality.

## 1 Introduction

Ground- and space-based measurement platforms are essential tools for continuous monitoring and evaluation of global, regional, and local air quality. Ground-based measurement networks are relatively inexpensive to deploy, are capable of comprehensive measurements of aerosol and trace gas chemistry and physial properties, and possess excellent temporal resolution.



However, these networks are sparsely distributed and often lack broad spatial coverage. Current satellite-based remote sensors, meanwhile, offer much greater horizontal spatial coverage but lack temporal resolution and specificity with regard to aerosol vertical extent and composition. While the temporal resolution of the space-based sensors is expected to improve dramatically with the anticipated launch of the TEMPO, GEMS, and Sentinel geostationary satellites over the next decade, the passive-based

sensors on these platforms continue to lack the vertically-resolved aerosol compositional information needed to constrain air quality and climate models. High-Spectral Resolution Lidar (HSRL) may be an attractive solution to this problem (Crumeyrolle et al., 2014).

In particular, the use of multiple wavelengths within an advanced HSRL system provides both the vertically-resolved location and spectrally-dependent optical properties of the aerosol. One such advanced HSRL is the airborne, three-wavelength HSRL-

2 system developed by NASA Langley Research Center. HSRL-2 measures aerosol backscatter coefficient ($\beta$) at 355, 532, and 1064 nm wavelengths and aerosol extinction coefficient ($\alpha$) at 355 and 532 nm wavelengths. As such, the instrument dataset is commonly referred to as $3\beta+2\alpha$. The spectrally-dependent optical properties ($3\beta+2\alpha$) detected by HSRL-2 encode information about aerosol size and chemical composition. Müller et al. (2014) and Chemyakin et al. (2014) have developed automated algorithm for retrieving aerosol physical properties (e.g., number, surface area, and volume concentration, as well

as effective radius) and chemical properties (e.g., real and imaginary refractive indices) from these signals.

The $3\beta+2\alpha$ retrieval technique has been employed by numerous studies in the past decade (Müller et al., 2001, 2003; Alados-Arboledas et al., 2011; Navas-Guzmán et al., 2013; Nicolae et al., 2013; Sawamura et al., 2014) and compares favorably to column-integrated retrievals from the Aerosol Robotic Network (AERONET) (Holben et al., 1998; Veselovskii et al., 2009; Sawamura et al., 2014). Yet, validation of the $3\beta+2\alpha$ retrieval against collocated, airborne profiling in situ measurements has

only been attempted for a limited case of continental pollution outflow described by Müller et al. (2014). Here, we extend these validation efforts to the DISCOVER-AQ study areas, which are ideal places to assess the skill of the HSRL-2 $3\beta+2\alpha$ retrieval and its relevance to air quality and PM$_{2.5}$.

A key challenge in relating the HSRL-2 and in situ aerosol measurements is that the former are made under ambient relative humidity (RH) conditions, while the latter are made under dry RH conditions (typically < 20%RH) (Zieger et al., 2011, 2012;

Sawamura et al., 2014). At high RH, hygroscopic aerosols uptake water, which affects their optically-relevant properties (e.g., size, morphology, and refractive index). Previous studies comparing HSRL and in situ extinction profiles have either neglected the influence of hygroscopic growth (e.g., Müller et al., 2014; Sawamura et al., 2014) or explicitly corrected the dry in situ data using measured f(RH) (e.g., Ziemba et al., 2013; Rogers et al., 2009).

In this work, we first convert the dry aerosol properties measured in situ to ambient conditions using a hygroscopic growth

adjustment, and then compare these data to those obtained via the HSRL-2 $3\beta+2\alpha$ retrieval in order to validate the retrieval technique in two very different environments: the wintertime San Joaquin Valley, California, and summertime Houston, Texas. Both regions experience periods of significant adverse air quality near the surface, but differ with regard to boundary layer dynamics, ambient temperature and humidity, and the presence/absence of episodic aerosol layers aloft in the free troposphere that may confound air quality inferences from column-integrated satellite retrievals such as aerosol optical depth (AOD). Our





findings inform the development of the next generation of airborne and spaceborne active remote sensors and efforts to use these advanced sensors to constrain the surface aerosol burden and aerosol properties relevant for air quality.

## 2   Field campaign description

The NASA DISCOVER-AQ field campaign (http://discover-aq.larc.nasa.gov; hereafter DAQ) was a four-year, four-city flight
campaign to understand the capabilities of satellite observations for diagnosing near-surface conditions relating to air quality. DAQ employed two aircraft with complementary flight patterns: 1) a high-altitude "satellite simulator" B-200 aircraft with HSRL-2, and 2) a profiling "in situ" P-3B aircraft measuring the vertical distribution of aerosol and trace gas properties within a series of several spiral points. The aircraft data were supplemented by ground-based instrumentation located below each spiral point consisting of surface monitors and upward-looking remote sensors. Here, we confine our analysis to data from the
2013 deployments in the San Joaquin Valley, California (DAQ CA) in January-February and Texas (DAQ TX) in September.

Figure 1 shows maps of the B-200 and P-3B aircraft flight tracks during DAQ California and Texas, as well as the designated spiral points. The NASA Langley B-200 King Aircraft (blue dashed line) completed two circuits per flight day at approximately 8.5 km altitude, while the NASA Wallops P-3B (solid red line) spiraled up and down over each label point (between approximately 0.015-0.5 and 5 km altitude) completing three circuits per flight day. The two aircraft followed coordinated flight tracks,
both flying over the designated air quality monitoring ground stations with close time coincidence, allowing for colocation of measurements from the in situ instruments suite and the HSRL-2.

## 3   Airborne high spectral resolution lidar

The NASA Langley airborne HSRL instrument technique has been described elsewhere (Hair et al., 2008), so only a brief description of the HSRL-2 instrument is given here. HSRL-2 measures aerosol backscatter coefficient ($\beta$) and depolarization
ratio ($\delta$) at 355, 532, and 1064 nm wavelengths following Fernald (1984) and aerosol extinction coefficient ($\alpha$) at 355 and 532 nm wavelengths following Shipley et al. (1983). Data are sampled at 2 Hz and 15-meter vertical resolution, which are then horizontally averaged for 10 seconds ($\beta$ and $\delta$) and 60 seconds ($\alpha$), which correspond to spatial averages of 1 and 6 km at nominal aircraft speed, respectively.

Aerosol microphysical properties are retrieved using an automated $3\beta + 2\alpha$ algorithm (Müller et al., 1999a; Veselovskii
et al., 2002; Müller et al., 2014) at a vertical resolution of 150 m during DAQ Texas and 75 m during DAQ California, based on differences in observed boundary layer heights during each campaign. The algorithm solves the inverse problem by representing the aerosol size distribution as a linear combination of eight, logarithmically equidistant triangular-shaped basis functions within an inversion window spanning particle radii of 0.03-8 $\mu$m (Müller et al., 1999a, b). The real part of the refractive index is allowed to vary from 1.325-1.8, while the imaginary part of the refractive index is allowed to vary from
0-0.1. For each $3\beta + 2\alpha$ set the algorithm is run 9 times. In 8 of those runs, the extinction and backscatter coefficents input are distorted by their respective uncertainties in different combinations in order to simulate possible measurement error scenarios



(see Table S1 in the supplemental material). Another run is performed with error-free input data. Hundreds of thousands of solutions are obtained with those 9 runs, and the 500 solutions with the lowest discrepancies are averaged and stored as the final solution (Müller et al., 1999b; Veselovskii et al., 2002). The standard deviation of the 500 best solutions provides a measure of the retrieval uncertainty, and only $3\beta + 2\alpha$ sets with uncertainties below 20% are used for the inversion (Müller et al., 1999b). The uncertainties originate from the HSRL-2 system's random noise and are estimated using the noise scale factor methodology of Liu et al. (2006). The $3\beta + 2\alpha$ algorithm allows for the differentiation of two modes: sub-micron fine mode and a super-micron coarse mode. Here, we examine only the fine mode retrieval as it is most directly comparable to the upper size limit of the aircraft in situ sampling inlet and particle sizing instrumentation.

## 4  Airborne in situ measurements

The aerosol dry size distribution, scattering coefficient, and hygroscopicity are measured via in situ sampling through an isokinetic, low-turbulence inlet mounted on the port side of the P-3B aircraft. The inlet transmits particles smaller than 5 $\mu$m diameter with greater than 50% efficiency (McNaughton et al., 2007). Dry and humidified scattering coefficients (450, 550, and 700 nm wavelengths) are measured with a pair of integrating nephelometers (Model 3563, TSI, Inc., Shoreview, MN, USA) (Pilat and Charlson, 1966; Clarke et al., 2002; Ziemba et al., 2013). The measurements are corrected for truncation errors following Anderson and Ogren (1998), while the scattering Ångström exponent is used to adjust the 550 nm scattering to 532 nm. One of the nephelometers is operated at dry relative humidity (RH$_{dry} \sim 10\%$), while the other is operated at elevated relative humidity (RH$_{wet} \sim 80\text{-}85\%$). The aerosol hygroscopicity, $\gamma$, is then

$$\gamma = \frac{\ln\left(\frac{\sigma_{\text{scat,wet}}}{\sigma_{\text{scat,dry}}}\right)}{\ln\left(\frac{100 - \text{RH}_{\text{dry}}}{100 - \text{RH}_{\text{wet}}}\right)} \tag{1}$$

where $\sigma_{scat,dry}$ and $\sigma_{scat,wet}$ are the aerosol light scattering coefficients under dry (RH$_{dry}$) and elevated RH conditions (RH$_{wet}$), respectively.

The ambient RH outside the aircraft is computed using the aircraft static temperature measurement and water vapor concentration measured by an open-path diode laser hygrometer (Diskin et al., 2002), and the ambient aerosol scattering coefficient $\sigma_{scat,amb}$ at this RH$_{amb}$ is determined as

$$\sigma_{\text{scat,amb}}(\text{RH}_{\text{amb}}) = \sigma_{\text{scat,dry}} \left[\frac{100 - \text{RH}_{\text{dry}}}{100 - \text{RH}_{\text{amb}}}\right]^{\gamma} \tag{2}$$

The atmospheric relevance of the $\gamma$ equation functional form has been recently been called into question and other multi-parameter methods have been suggested (Kotchenruther et al., 1999; Carrico et al., 2003; Brock et al., 2016; Orozco et al., 2016). In particular, the smooth nature of the $\gamma$ parameterization may overestimate the scattering of effloresced dry particles at RHs below their deliquescence point (e.g., pure ammonium sulfate), or by as much as 20% for 50-90% RH based on multi-RH




fits (Brock et al., 2016). Other studies have shown the $\gamma$ parameterization to be suitable for describing the hydration of sea salt, for example (Zieger et al., 2010). As will be discussed in Section 6, the hygroscopically-adjusted in situ measurements tend to underestimate the HSRL-2 extinction and scattering, which is opposite of what would be expected to result from the error associated with the $\gamma$ parameterization under the moderate RH levels observed during DAQ.

Dry aerosol absorption coefficient (470, 532, and 660 nm wavelengths) is obtained from a Particle Soot Absorption Photometer (PSAP; Radiance Research, Shoreline, WA, USA) that was heated to 30°C to prevent water condensation on the filter substrate. PSAP measurements are corrected for filter artifacts following Virkkula (2010).

Aerosol extinction coefficients are computed as the sum of the scattering and absorption coefficients, neglecting hydration effects on the absorption coefficient, which are highly uncertain and likely to be minimal for the largely non-absorbing aerosols
observed during most of DAQ (Flores et al., 2012). The average single scattering albedo values at 550 nm (dry conditions) during DAQ CA and TX were $0.96 \pm 0.03$ and $0.91 \pm 0.04$, respectively.

The primary aerosol dry size distribution measurement is from an Ultra-High Sensitivity Aerosol Spectrometer (UHSAS; Droplet Measurement Technologies, Inc., Boulder, CO, USA) measuring 0.06 $\mu$m to 1 $\mu$m diameter particles. To evaluate the impact of coarse mode particles, a Laser Aerosol Spectrometer (LAS Model 3340; TSI, Inc.), measuring 0.09 $\mu$m to 7.5 $\mu$m
diameter particles, was also flown during the DAQ CA campaign. The upper limit of the LAS is limited by the roughly 5 $\mu$m cutoff size of the aircraft inlet. Both instrument size calibrations were field calibrated with both NIST-traceable polystyrene latex spheres and size-classified ammonium sulfate aerosols. Here, we use the ammonium sulfate size calibrations since the refractive index of ammonium sulfate aerosol is closer to that of most atmospheric particles.

## 5  Methodology

Most of the in situ data used in this study were obtained at dry conditions except for the scattering at 550 nm which was also measured at wet conditions. Therefore, in order to properly compare the in situ measurements to the HSRL-2 measurements (and retrievals), it was necessary to adjust the dry in situ measurements to account for hygroscopic effects at the ambient RH.

For spherical particles, optical properties like scattering and absorption coefficients can be calculated with Mie theory if the size distribution and the complex refractive index ($m$) of the aerosol particles are known. In this study we use the measurements
of scattering and absorption coefficients and size distributions at dry conditions to retrieve $m_{dry}$ values. Once $m_{dry}$ is retrieved, it is possible to use it with its respective size distribution in a hygroscopic growth model to reproduce the scattering coefficient measured at ambient conditions. In this process we are able to infer the effective hygroscopic growth factor, $\bar{g}$.

In the following subsections we describe how profiles of $m_{dry}$, $m_{amb}$ and $\bar{g}$ are retrieved from the in situ measurements.

### 5.1  Data selection

Co-incident vertical profiles of HSRL-2 data products within a 10 km radius of each spiral location and within 30 minutes are matched to the in situ spiral data. A total of 172 sets of coincident profiles (i.e. in situ and HSRL-2) were considered for the analysis of the microphysical properties: 95 from DAQ TX and 77 from DAQ CA. Data were screened to remove any periods





where the HSRL-2 depolarization ratio at 532 nm wavelength ($\delta_{532}$) was greater than 5%, which is indicative of aspherical particles such as dust. This screening step is important because the $3\beta+2\alpha$ retrieval algorithm applies Mie theory (Mie, 1908), which is applicable only to spherical particles. This screening step has a larger impact on the DAQ CA data set than on the DAQ TX data set because of the shallow wintertime boundary layers observed in California. Ultimately, 108 profiles had valid

data points for both HSRL-2 retrievals and adjusted in situ measurements: 76 from DAQ TX (630 data points) and 32 from DAQ CA (126 data points).

The P-3B spiral diameters were 6-10 km with an average vertical resolution of 5 m. Consequently, the higher resolution, 1 Hz in situ data were averaged to match the HSRL-2 data vertical resolution. The DAQ data used in this study are publicly available at *http://www-air.larc.nasa.gov/missions/discover-aq/discover-aq.html*.

## 5.2   Part I: Retrieval of dry refractive index

The first step in our analysis is to apply the measured in situ dry size distribution, dry scattering coefficient, and dry absorption coefficient to derive the aerosol dry refractive index ($m_{dry}$) using Mie theory. This iterative procedure is summarized in the upper portion of Figure 2. Measurement inputs are shown as blue text and computed outputs as green text. The model is solved iteratively over the range of $m_{dry}$ = (1.33 to 1.7) - (0 to 0.03)$i$ to match the measured dry aerosol optical properties, and the final

derived dry refractive index is the average of all solutions that meet the convergence criteria. Example profile outputs are shown in the lower portion of Figure 2 for a spiral over Channel View, in Houston, TX, on September $11^{th}$, 2013 between 21:06 - 21:15 UTC. Red points indicate the Mie theory converged solution and show good agreement with the measured scattering and absorption coefficients. We assume that the aerosol refractive index is wavelength-independent over the spectral range covered by the in situ instruments (i.e. 450 nm to 700 nm). This assumption is also consistent with the lidar retrieval methodology.

## 5.3   Part II: Retrieval of effective growth factors

The next step is to compute the aerosol hygroscopic growth due to RH-dependent water uptake, and employ Mie theory to determine the aerosol optical properties under ambient RH conditions. The hygroscopic growth factor, $g(\mathrm{RH}, \mathrm{D_{dry}})$ is defined as the ratio of the diameter of the hydrated particle at a given RH to its dry diameter. Again, we assume that the aerosol population is internally mixed with size-invariant composition, which means that we can represent the aerosol using a single,

effective hygroscopic growth factor, $\bar{g}$, similar to Zieger et al. (2010). Thus, the entire size distribution shifts toward larger diameters by the same factor of $\bar{g}$ under elevated RH conditions.

As the particle takes on water, its refractive index decreases from the dry particle $m_{dry}$ toward that of pure water ($m_{H_2O}$ = $1.33 \pm 0i$, Hale and Querry (1973)). We assume that the aerosol is internally mixed and that the hydrated particle refractive index ($m_{amb}$) is calculated as a volume-weighted average of $m_{dry}$ and $m_{H_2O}$ as

$$m_{amb} = \frac{m_{dry} + m_{H_2O}(\bar{g}^3 - 1)}{\bar{g}^3} \tag{3}$$





The dry size distribution and dry refractive index computed in Section 5.2 are combined with the ambient aerosol scattering coefficient (Equation 2) to iteratively determine $\bar{g}$. This method is depicted in Figure 3 with the model input shown as blue text and model outputs shown as green text. The algorithm iterates over $\bar{g}$ = 1 to 2 until the Mie theory computed ambient scattering coefficient is within 1% of that computed from the in situ data via Equation 2. Profiles of the ambient scattering coefficient,

effective hygroscopic growth factor, and ambient complex refractive index are shown in the lower portion of Figure 3.

In addition to $\bar{g}$, the aerosol hygroscopicity is often represented as a single parameter $\kappa$ following Petters and Kreidenweis (2007) with soluble salts exhibiting higher values of $\kappa$ ($> 0.6$) and organic species exhibiting $\kappa$ values $< 0.3$. A wettable but insoluble aerosol particle would have a $\kappa$ of 0. Figure 4a shows the distribution of $\bar{g}$ computed assuming an ambient RH of 85% for all DAQ California and Texas data points. Lines of constant $\kappa$ are shown for comparison. While we expect that the retrieved

$\bar{g}$ represents the dominant aerosol hygroscopicity of the population, the method for calculating $\bar{g}$ may obscure less hygroscopic modes which reflect a minor contribution to the overall scattering. Overall, the aerosol encountered in Texas appears to be more hygroscopic than in California, with the former consistent with organic-sulfate mixtures and the latter more consistent with a more dominant organic aerosol fraction.

### 5.4 Part III: Closure study: Optical properties evaluation

While the in situ ambient $\alpha_{532}$ (derived in Section 4) is directly comparable to the HSRL-derived $\alpha_{532}$, the in situ hemispheric scattering coefficient must first be adjusted using an angular-dependent scattering phase function. Here, we again invoke Mie theory combined with the aerosol dry size distribution, $\bar{g}$, and $m_{amb}$ found in Section 5.3 to compute the aerosol backscatter coefficient ($\beta_{532}$) from the in situ data. Similarly, this Mie theory model is applied to calculate the $\alpha_{355}$, $\beta_{355}$, and $\beta_{1064}$. This facilitates comparison between the HSRL-2 and in situ data in two ways. First, we examine the skill of the HSRL-2

microphysical retrieval of aerosol number, surface area, and volume concentration as well as effective radius against the in situ measurements. Second, we examine the applicability of the Mie theory model and in situ data for calculating the ambient aerosol optical properties measured by HSRL-2.

## 6   Results and Discussion

### 6.1 Retrieved Aerosol Microphysical Properties

Example profile-to-profile comparisons from DAQ TX of aerosol microphysical properties are presented in Figure 5, where the blue squares are the hygroscopically-adjusted in situ measurements and the gray circles are the HSRL-2 $3\beta + 2\alpha$ retrievals. The error bars represent the HSRL-2 retrieval uncertainties (see Section 3). In situ error bars are not shown for clarity, but are estimated to be approximately 10-15%. The $3\beta + 2\alpha$ retrievals are in excellent agreement with the in situ data and capture the vertical structure of both the boundary layer aerosols as well as a layer aloft at 3 km near Smith Point on 9/12.

Figure 6 shows the complete comparison of all 108 coincident points obtained with the HSRL-2 and the in situ instruments during DAQ CA and TX. Volume and surface area concentrations present the best correlations with correlation coefficients





(r) of 0.73 and 0.67, respectively for California and 0.75 and 0.74 in Texas. The median biases between the retrieval and in situ data in California and Texas, respectively, are: 47% and 33% for number concentration, 25% and 15% for surface-area concentration, 3% and 7% for volume concentration, and -25% and -7% for effective radius (Figure 7).

The bias between the HSRL-2 retrieval and in situ data is also computed neglecting aerosol hydration effects (light blue boxes in Figure 7). While there is no effect on the retrieved aerosol number, the decreased aerosol size results in much greater biases for particle effective radius, surface area, and volume concentrations. The largest impact is on the retrieved aerosol volume since it is the third moment of the size distribution. (Brock et al., 2016) showed that representing aerosol hygroscopic effects on light scattering with a $\gamma$ parameterization may overestimate light scattering by as much as 20% over the range of 50-90% RH commonly observed during DAQ TX (orange bars in Figure 4b ). Potential uncertainties associated with aerosol water uptake during DAQ CA are expected to similar or even less, owing to the drier conditions (<60%) that were generally encountered (blue bars in Figure 4b). However, overprediction of aerosol hygroscopic growth cannot explain the biases shown in Figure 7, where the HSRL-2 retrieval approximately equals or exceeds the in situ measurements.

## 6.2  Optical closure study

In addition to comparing the retrieved microphysical properties to the in situ measurements, we also assess the applicability of Mie theory to predict the ambient aerosol optical properties from the in situ measurements. This comparison is shown in Figure 8. A greater number of data points are available for comparison owing to the higher vertical resolution of the HSRL-2 optical data products (15 m resolution ) as compared to the microphysical retrievals (75-150 m resolution). The results for California are displayed on the top row and the results for Texas on the bottom row. The red dashed lines in Figure 8 correspond to bisector linear regressions and their fit parameters are given in Table 1. The measured aerosol optical properties measured by HSRL-2 are well correlated with the Mie theory calculations; however the HSRL-2 measurements are significantly larger than the calculated values. This deviation is accentuated in the Texas dataset.

We note that unlike the HSRL-2 $3\beta+2\alpha$ retrieval, which was restricted to the submicron aerosol mode for our comparison, the HSRL-2 extinction and backscatter coefficients shown in Figure 8 are not constrained. Consequently, scattering and extinction from coarse aerosol mode may help to explain this systematic underestimation of the in situ data, since these aerosols are too large to be efficiently transmitted through the aircraft inlet or are too large to be sized by the UHSAS .

Comparison between the column-integrated HSRL-2 extinction and aerosol optical thickness (AOT) from ground-based, DAQ AERONET measurements (co-incident to within 2.5 km and 10 minutes) show excellent agreement (Figure S1). Meanwhile, the calculated AOT using the aircraft in situ ambient extinction coefficients also underestimates the AERONET AOT (Figure 9). For Texas the median [interquartile] ratio of in situ AOT to AERONET is 0.60 [0.45, 0.67], which increases with increasing AERONET Ångström exponent (See Figure S2). This trend supports the idea that the coarse aerosol mode is the cause of the difference between HSRL-2 and in situ scattering and extinction coefficients.

For California the median [interquartile] in situ AOT to AERONET ratio is similar: 0.61 [0.51, 0.79]. However, there is no clear correlation with the Ångström exponent. Rather, the discrepancy is likely due to near-surface aerosol that was below the minimum aircraft flight altitude. For DAQ CA, the median vertical profile of aerosol extinction shows the aerosol to be





concentrated in the first 500-1000 m altitude, for DAQ TX, the aerosol extend up to 3-4 km altitude (Figure S3). To try to sample these near surface aerosols, the P-3B performed missed approaches at local airports located near some, but not all, of the spiral locations. The missed approaches in California show that these measurements are especially important for the calculation of the AOT. If the missed approach legs are removed from the data set, the median in situ : AERONET AOT ratio decreases from 0.61 to 0.38 [0.30, 0.44].

Figure 11A shows the average size distribution retrievals (in volume) from AERONET/DRAGON sunphotometers obtained during DAQ 2013. Figure 11B shows that the median aerosol optical depth (AOD) fine fraction (O'Neill et al., 2001) is smaller for Texas (0.73) than for California (0.79). The AOD fine fraction observed in Texas shows larger variability toward lower fractions, further supporting the hypothesis that undersampling of larger particles in Texas was a factor in the discrepancies of measured and calculated $3\beta + 2\alpha$. Kassianov et al. (2012) highlight the importance of sampling supermicron particles due to the potential of significant impact of supermicron aerosols in the calculation of aerosol radiative properties in areas where the relative contribution of the supermicron fraction is often overlooked.

To further examine the influence of coarse mode aerosol on the calculated in situ optical properties, we reran the Mie theory calculations using size distributions from the LAS (0.09 - 5$\mu$m diameter) instead of the UHSAS (0.09 - 5$\mu$m diameter). This analysis was carried out only for DAQ CA, as LAS data are not available for DAQ TX. Using the LAS size distributions reduces the bias by a modest amount (7% on average) for all but the 1064 nm scattering coefficient, which improved by 17%.

The median biases between HSRL-2 and in situ calculated aerosol backscatter and extinction coefficients are shown in Figure 10. In general, better agreement can be seen for the extinction coefficients than for the backscatter coefficients.

## 7   Summary and Conclusions

We present an extensive evaluation of aerosol microphysical retrievals obtained from an advanced, multi-wavelength HSRL in different environments relevant to U.S. air quality. Coincident lidar $3\beta + 2\alpha$ profile retrievals and in situ vertical profiles are compared with respect to aerosol fine mode microphysical properties. The aerosol effective radius, surface area and volume concentration retrieval products are in excellent agreement with in situ measurements, once the latter have been corrected for hygroscopic water uptake. Meanwhile, the retrieval of fine mode aerosol number is much less constrained, but still within roughly 50% of the in situ values. The best agreement is observed for fine mode aerosol volume concentrations, arguably the most important for assessing $PM_{2.5}$ and air quality, where the median biases between the retrieval and in situ measurements were only 3% and 7% for California and Texas, respectively.

Comparison of the HSRL-2 measurements of aerosol scattering and extinction coefficients with predictions from Mie theory and in situ data show larger biases. This is attributed to the presence of coarse mode aerosol that are detected by HSRL-2 but that are too large to be well sampled by the in situ instrumentation. Integrating HSRL-2 extinction over the vertical profile yields aerosol optical thicknesses (AOTs) that agree well with ground-based AERONET measurements, which lends support to this explanation. Similarly, using the LAS to extend the upper size range of the DAQ CA in situ aerosol data from 1 $\mu$m to $\sim$5$\mu$m reduces the negative bias of the in situ calculations by 7%, and comparison of in situ and AERONET-derived AOT





shows a dependence on Ångström Exponent (a proxy for aerosol size). These findings emphasize the need to validate remote sensor retrieval algorithms that are capable of differentiating sub- and super-micron aerosol modes, since only the former are currently readily sampled by aircraft in situ instruments.

The NASA DISCOVER-AQ data set is ideal for assessing the performance of advanced HSRL aerosol microphysical re-
5  trievals and should be used to evaluate future retrieval schemes using the same $3\beta + 2\alpha$ data that we have applied here. Such methods include optimal estimation frameworks and the "arrange-and-average" technique of Chemyakin et al. (2014). Together with such future contributions, this study helps us to better understand the robustness and limitations of advanced, multi-wavelength lidar retrievals and their applicability to constraining air quality from space.

*Acknowledgements.* This research was supported by an appointment to the NASA Postdoctoral Program at the NASA Langley Research
10  Center, administered by Oak Ridge Associated Universities (now administered by Universities Space Research Association) under contract with NASA. The authors would like to thank the flight crew from both NASA B200 and P-3B, and the team members of the NASA Langley Aerosol Research Group Experiment (LARGE) and others who contributed to the data processing and archival process for the DISCOVER-AQ data. The authors would also like to thank Amy Jo Scarino for her contribution with the AOT comparison between HSRL-2 and AERONET measurements during DAQ CA and TX.



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




**Table 1.** Parameters of bisector linear regression fit between the HSRL-2 measurements (x) and the reconstructed version of $3\beta+2\alpha$ (y: from in-situ measurements). The columns under UHSAS contain the parameters from the linear regressions depicted as red dashed lines in Figure 8. The column under LAS contains the parameters from the linear regressions obtained when the reconstructed version (y) is calculated using LAS measurements instead of UHSAS measurements (not shown, see text). R refers to the Pearson's correlation coefficient. Extinction coefficient ($\alpha$) unit is $\mathrm{Mm}^{-1}$ and backscattering coefficient ($\beta$) unit is $\mathrm{Mm}^{-1}\,\mathrm{sr}^{-1}$

| | UHSAS | | LAS |
|---|---|---|---|
| | DAQ CA | DAQ TX | DAQ CA |
| $\alpha_{355}$ | $y = 0.77x - 23.2, R = 0.89$ | $y = 0.66x + 7.6, R = 0.88$ | $y = 0.84x - 26.5, R = 0.90$ |
| $\alpha_{532}$ | $y = 0.94x - 20.4, R = 0.88$ | $y = 0.67x + 4.1, R = 0.86$ | $y = 0.98x - 19.4, R = 0.91$ |
| $\beta_{355}$ | $y = 1.13x - 1.7, R = 0.88$ | $y = 0.57x - 0.1, R = 0.72$ | $y = 0.98x - 1.1, R = 0.89$ |
| $\beta_{532}$ | $y = 1.26x - 1.4, R = 0.86$ | $y = 0.55x - 0.01, R = 0.79$ | $y = 1.08x - 0.96, R = 0.88$ |
| $\beta_{1064}$ | $y = 0.81x - 0.34, R = 0.83$ | $y = 0.51x + 0.03, R = 0.46$ | $y = 1.02x - 0.40, R = 0.80$ |

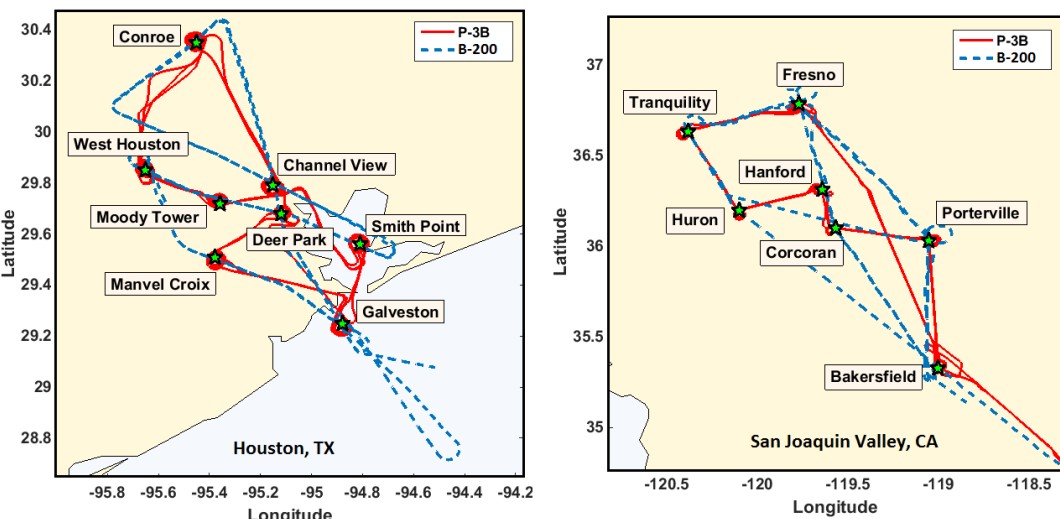

**Figure 1.** Map of the flight tracks for DAQ California (right) and DAQ Texas (left). The green stars mark the location of the ground stations over which the P3B (red solid line) spiraled. The HSRL-2 instrument onboard the King Air (blue dashed line) flew over the same ground stations.



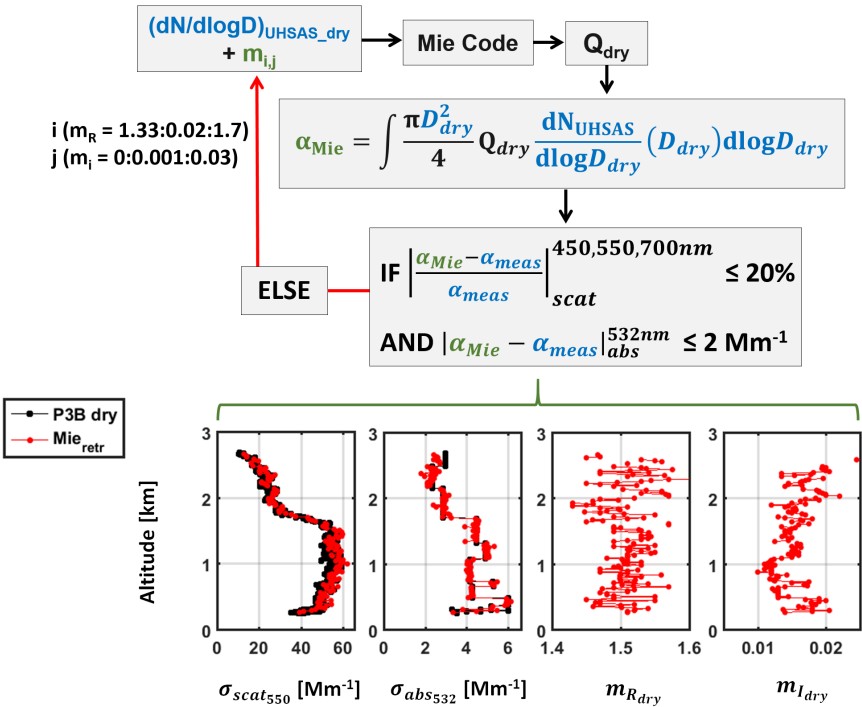

**Figure 2.** Diagram representation of Part I in which the dry complex refractive index pairs ($m_{R_{dry}}$, $m_{I_{dry}}$) are retrieved from the in situ measurements. UHSAS measurements of size distributions are obtained for particles with D < 1 $\mu$m while the scattering and absorption measurements were obtained for particles with D < 5$\mu$m. The blue terms in the block diagram refer to measured quantities. $Q_{dry}$ refers to the scattering or absorption efficiencies calculated with the Mie code. The profiles shown at the bottom of this figure were obtained over Channel View on September 11$^{th}$, 2013 between 21:06 - 21:15 UTC, during DAQ TX. The plots show the comparison between the in situ measurements ("P3B dry" in black) and the retrieved ("Mie$_{retr}$") dry scattering at 550 nm ($\sigma_{scat_{550}}$), and absorption at 532 nm ($\sigma_{abs_{532}}$) obtained with the complex refractive index retrieved in this step.





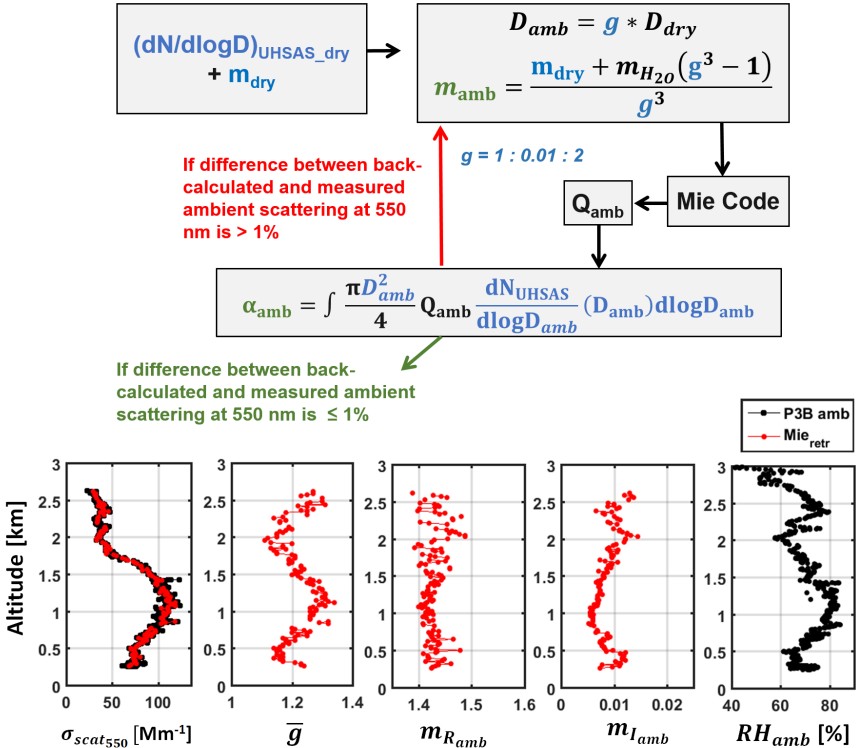

**Figure 3.** Diagram representation of Part II in which effective growth factor $\bar{g}$ and ambient complex refractive index $m_{amb}$ are retrieved using in situ data. The profiles shown at the bottom of this figure were obtained over the Houston area, on September $11^{th}$, 2013 between 21:06 - 21:15 UTC, during DAQ TX. Measurements are shown in black and retrievals in red in the bottom plots.

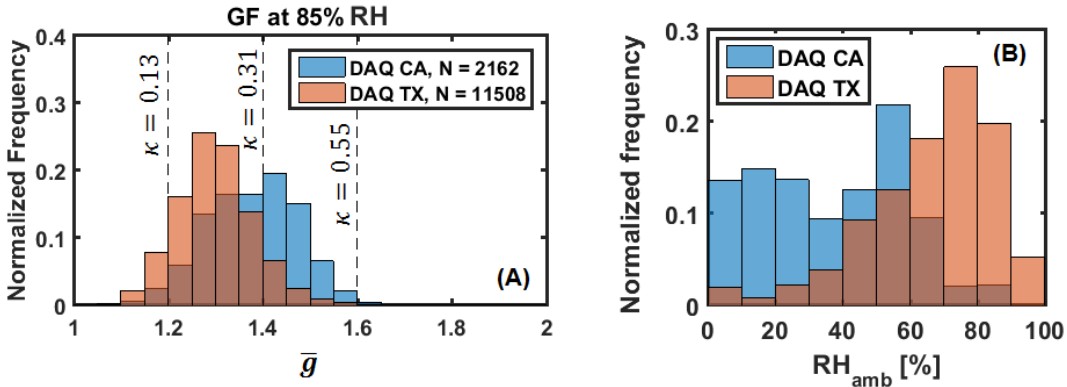

**Figure 4.** (A) Distribution of retrieved $\bar{g}$ values at RH = 85%. Median $\bar{g}$ values were 1.38 in California and 1.29 in Texas. (B) Relative humidity measurements during DAQ CA and TX.





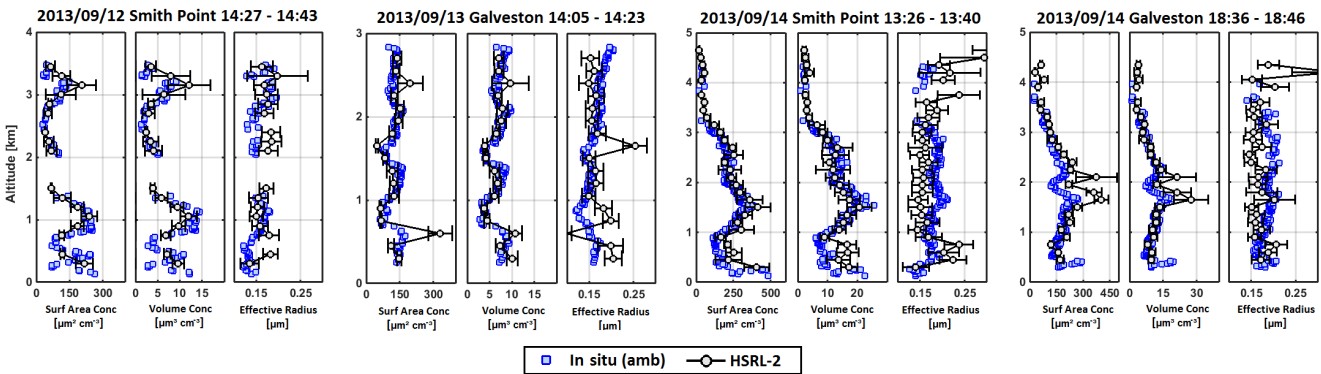

**Figure 5.** Examples of profile-to-profile comparisons between HSRL-2 fine mode retrievals and in situ measurements (corrected for ambient RH) of surface-area and volume concentrations, and effective radius obtained during DAQ TX. The error bars represent the uncertainties of the HSRL-2 retrieval.

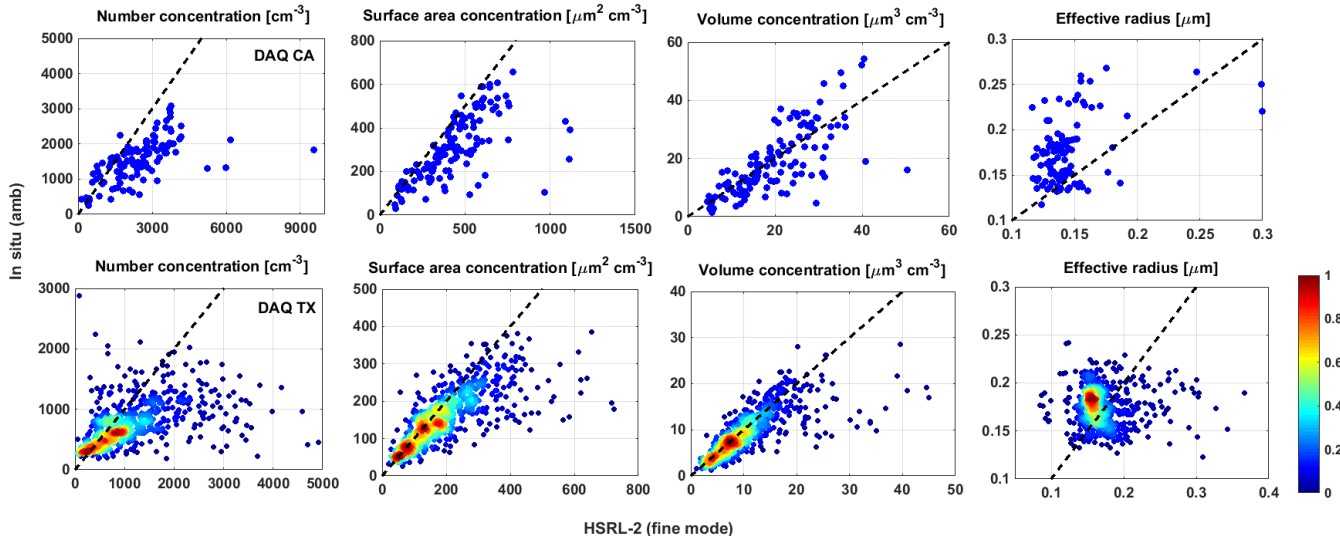

**Figure 6.** Comparison of number, surface-area and volume concentrations, and effective radius (fine mode only) for all coincident points between HSRL-2 (x-axis) and in situ (corrected for ambient RH, y-axis) obtained during DAQ CA and TX. The black dashed lines represent 1:1. There are 126 points in the DAQ CA plots and 630 data points in the DAQ TX plots. For the latter, due to the large number of overlapping points, the scatter plot is color-coded by the density of points. The colorbar is normalized to 1.





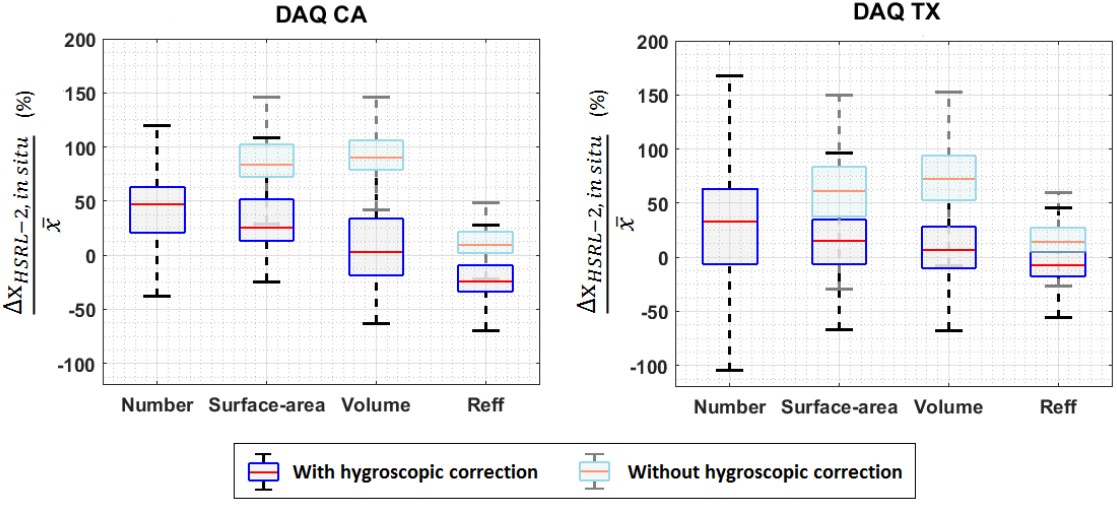

**Figure 7.** Statistics of the comparison between the HSRL-2 retrievals and the in situ measurements of number, surface-area, and volume concentrations, and effective radii for DAQ CA (left) and DAQ TX (right) with and without the hygroscopic correction applied to the in situ measurements (fine mode only). The boxplots represent the distribution of the biases observed between the HSRL-2 retrievals and the in situ measurements for each parameter. The relative bias was calculated as the ratio of the difference between HSRL-2 retrievals and the in situ measurements ($\Delta x$) to the average between the HSRL-2 retrievals and the in situ measurements ($\bar{x}$). The red line represents the median value, the boxes edges represent the $25^{th}$ and $75^{th}$ percentiles ($q_1$, $q_3$, respectively), and the whiskers represent the outlier boundaries ($q_1 - 1.5\times$ IQR) and ($q_1 + 1.5\times$ IQR), where IQR is the interquartile range, defined as IQR = $q_3$ - $q_1$.





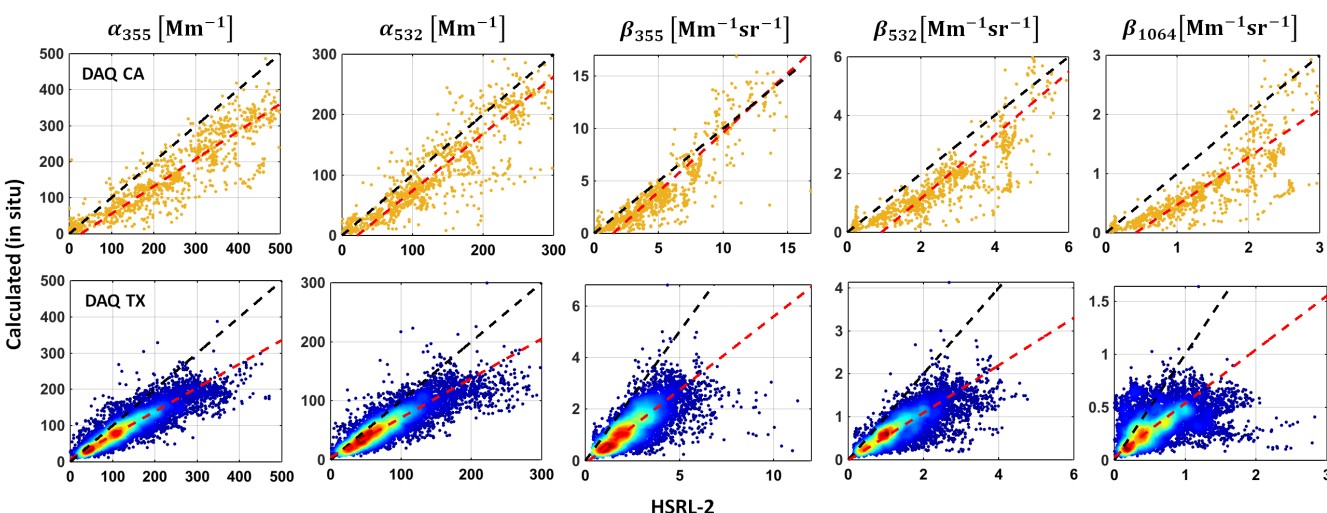

**Figure 8.** Comparison between calculated and measured $3\beta + 2\alpha$. The measurements obtained with the HSRL-2 are on the x-axis and the optical parameters calculated from the adjusted in situ size distributions measured with the UHSAS (see text). The black dashed lines represent the 1:1 line and the red dashed lines correspond to bisector linear regression fit of each dataset (see Table 1 for fit parameters). DAQ TX data are presented as scatter density plots.





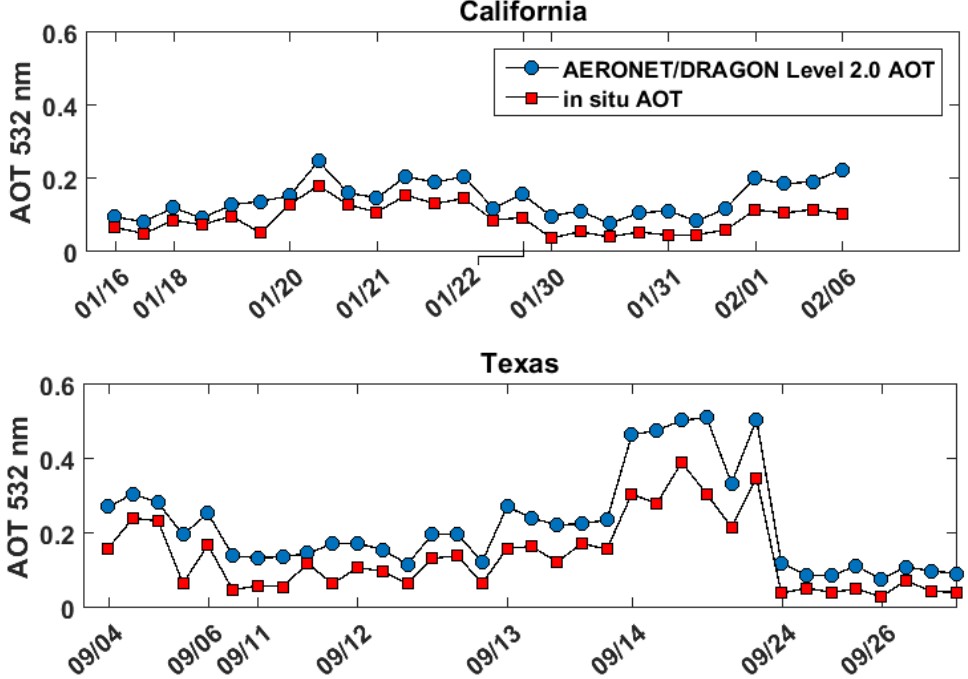

**Figure 9.** AERONET and in situ AOT measurements during DAQ CA and TX. In situ values were obtained by integration of the ambient extinction vertical profile.

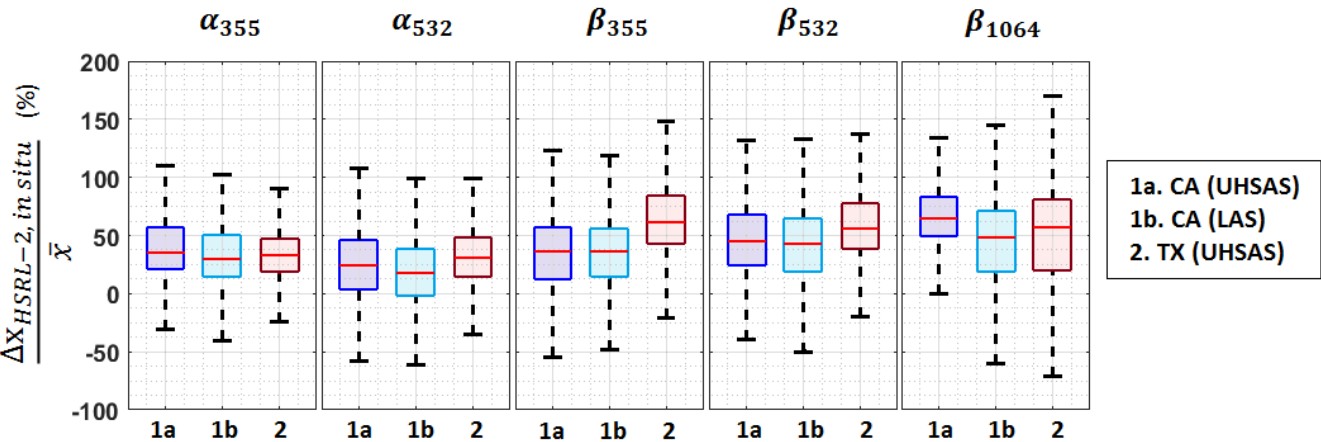

**Figure 10.** Statistics of bias observed between HSRL-2 measurements of extinction ($\alpha$) and backscatter coefficients ($\beta$) and those calculated from the adjusted in situ measurements and Mie theory for the DAQ CA data (using UHSAS and LAS) and for the DAQ TX data (UHSAS only). UHSAS measures the size distribution of sub-micron particles, while the LAS measures the size distribution of sub- and super-micron particles.




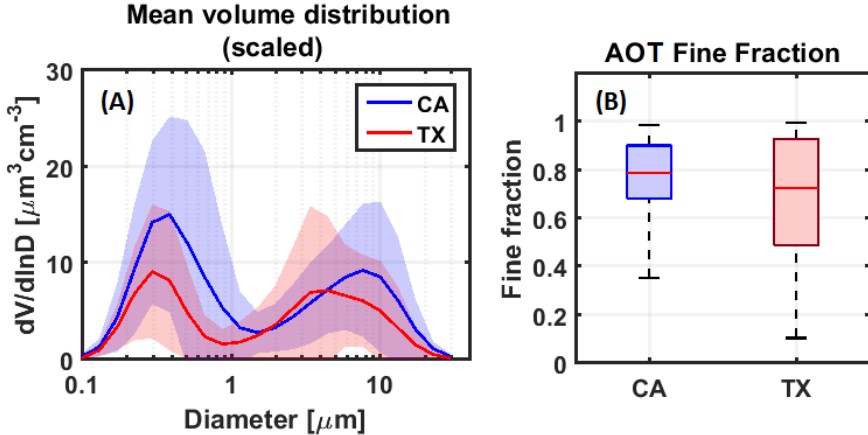

**Figure 11.** (A) Mean volume distribution retrieved from AERONET/DRAGON sunphotometer measurements, scaled to a maximum aerosol layer height of 1 km for CA and 3 km for Texas. (B) Statistics for the aerosol optical depth fine fraction.