# Peer review of "HSRL-2 aerosol optical measurements and microphysical retrievals vs. airborne in situ measurements during DISCOVER-AQ 2013: an intercomparison study"

_Atmospheric Chemistry and Physics, 2016_

## Referee Comment (RC1) · Anonymous Referee #1 · 14 Feb 2017

This work is a very important step towards space-borne observations of vertical distribution of aerosol microphysical properties. It evaluates airborne HSRL-derived aerosol number concentration, surface area concentration, volume concentration and effective radius against collocated airborne in-situ measurements during the DISCOVER-AQ campaign in Texas and in California. This work is worthy of publication, after minor revisions:

Page2, Line 16-19: "The (. . .) retrieval technique compares favorably to column-integrated retrieval (. . .)". The reader would benefit from quantitative results here.

[Figure]

Page2, Line 20: " (..) limited case of continental pollution outflow (. . .)". Please add location and season.

Page 4, line 25: consider replacing "equation functional form has been recently been" by "equation (2) was recently called into. . ."

Page 5, Line 16: "both" is repeated

Page 5, line 18: is there any reference for "refractive index of ammonium sulfate aerosol is closer to that of most atmospheric particles"?

Page 5, line 25: the authors might want to add that there were no airborne instruments retrieving refractive index (RI) during DISCOVER AQ (e.g. DASH-SP or PINeph during SEAC4RS) for a direct comparison to their HSRL-2-retrieved RI?

Page 6, Line 4: The connection between the shallow wintertime boundary layer in California and the larger impact of the spherical filter is not straightforward. The authors should explain/ expand more.

Page 6, Line 17 and Figure 2 & 3: Can the red points (i.e. "Mie retr") be changed to green, as they are retrieved outputs? I also suggests to (i) change the red arrows in black, (ii) change red text in black on figure 3, (iii) delete green arrow and text on fig 3 as this is implied by the red text above, (iv) change mi,j into (mRdry, mIdry) and delete i,j, (v) change Qdry and Qamb in green instead of black, (vi) add " Houston, TX" after "Channel View"

Page 6, Line 17: "good agreement" needs more quantification e.g. within how many percent?

Page 6, Line 24 and 28: "internally mixed" is repeated twice.

Page 7, line 25: "Examples of profile-to-profile"

Page 7, line 28: is it "10-15%" on all parameters?

Page 7, line 30: "108 coincident profiles (see section 5.1)" instead of "points"

Page 8, line 1: The reader would benefit from a table showing statistics, equivalent to Table 1 or at least add the correlation coefficients on Figure 6. Comparisons regarding number concentration and effective radius also need to be described in the text.

Page 8, line 10: "are expected to be similar" and "<60% RH"

Page 8, line 19: "measured" is repeated twice

Page 8, line 28: please add "vertical integration of ambient extinction coefficient"

Page 8, line 26 to Page 9, line 18: the authors use 6 figures (i.e. figure 8, S1, 9, S2, S3, 11) to explain that the coarse aerosol mode (as well as the presence of aerosols below the aircraft in California) is the cause of the difference between HSRL-2 and insitu scattering and extinction coefficients. The reasoning could be clarified and some figures could be merged together. For example, why not show the HSRL AOD together with AERONET and insitu AOD on Figure 9? The switching back and forth between supplementary and main figures is not obvious as well. Also Figure 10 should be described in much more detail and paired with the description of figure 8.

Figure 9, legend: add "vertical integration"

Figure 11, legend: "scaled to a maximum aerosol layer height of 1 km for CA.." needs more explanation and (B) should also read AERONET/DRAGON

Figure S2, legend: should read "ratio of insitu AOD to AERONET AOD"

Page 9, line 7: "AERONET" should be added after "median"

Page 9, Line 23: The authors need to be cautious: "excellent agreement" is a strong statement and does not apply to the HSRL-derived effective radius for example.

Page 9, line 25: "within roughly 50% of the insitu values". This does not seem to have been described in the text. Figure 6 needs to show +- 50% lines.

---

## Referee Comment (RC2) · Anonymous Referee #3 · 1 Apr 2017

This manuscript describes a very important and valuable study which compares the HSRL-2 retrieval algorithm to in situ airborne profiling measurements. This type of study is essential for the validation of the HSRL-2 algorithm and the understanding of its abilities/limitation. Also pointing out the need for in situ optical aerosol measurements at ambient conditions (RH and course mode aerosols). The paper is very well written and the plots are clear. The work is carefully done, including stuffiest statistics. I recommend publishing after addressing these comments:

General comments:

*P8 Line22-25: The authors provide an explanation for the discrepancy between the HSRL-2 and the calculated in situ, addressing the presents of course mode aerosols. The reviewer agrees this could be a possible explanation and the authors supports their explanation nicely using the AERONET data. However, it is not clear what type of aerosols do the authors suggest could explain this discrepancy, given that depolarization ratio at 532 nm wavelength greater than 5%,was screened ( meaning the authors have removed dust particles). Also why the Texas data has accentuated deviation compared to California? What should be the size of these course mode aerosols? If incorporating the $5\mu$m LAS data was not stuffiest to explain the discrepancy. Could there be any other explanation for this discrepancy?

*P6 Line 18 "We Assume that the aerosol RI is wavelength independent" (450-700nm) , this assumption requires a short justification, especially in the lower wavelength range, close to UV. It seems like a reasonable assumption for mostly scattering aerosols (as shown in this study), but needs to be mentioned. Were there any absorbing aerosols present in one of the sites? (e.g. BB aerosols)

*Was there any chemical information of the aerosols on the airplane platform or ground base that could support the conclusions?

*The authors report that the HSRL-2 vertical profile are products within 30 min matched to the spiral data, was the aerosol population well mixed, for this comparison to meaningful?

*P6 line 25: "The entire size distribution shifts towered larger diameters", it seems like reasonable assumption, depending on the type of the aerosol. What is the error associated with this assumption?

Specific comments:

*Suggest to make figure 6 more clear by moving the headline of DAQ-CA and the DAX TX outside (same comment for Figure 8)

*P8 Line 7: Please delete brackets from Brock et al., 2016

*Figure 7: suggest to modify Reff to: R_eff (subscript the eff)

*P9 line 14: "LAS (0.09 - 5$\mu$m diameter) instead of the UHSAS (0.09 - 5$\mu$m diameter)". There is a typo: the same range is reported for both instruments.

---

## Author Comment (AC1) · 8 May 2017

**Response to reviewers:**

The authors would like to thank both reviewers for their comments and suggestions. We addressed every comment and compiled the response to both reviewers in this single file. Some of the modifications are not explicitly quoted here. However, a new version of the manuscript highlighting the changes will be available. The modifications suggested by Reviewer #1 are highlighted in yellow and those suggested by Reviewer #3 in cyan.

**Anonymous Referee #1**

Page 2, Line 16-19: "The (…) retrieval technique compares favorably to column-integrated retrieval (…)". The reader would benefit from quantitative results here.

Done. The paragraph was modified per the reviewer: "*In Veselovskii et al. (2009) the agreement between the retrievals for the fine mode of volume concentrations and effective radius obtained from lidar and AERONET was within 14% and 22%, respectively. In Sawamura et al. (2014) the agreement was found to be within 13% and 6% for the same variables.*"

Page 2, Line 20: "(…) limited case of continental pollution outflow (…)". Please add location and season.

Done. Location and season were added: "*(…) limited case of continental pollution outflow from the northeastern coast of the US out over the western Atlantic Ocean during summer as described (…)*".

Page 4, Line 25: consider replacing "equation functional form has been recently been" by "equation (2) was recently called into…"

Done. The sentence was modified per the reviewer.

Page 5, Line 16: "both" is repeated.

Done. The sentence was modified per the reviewer: "The UHSAS and LAS instruments were field calibrated with both NIST- (…)".

Page 5, Line 18: is there any reference for "refractive index of ammonium sulfate aerosol is closer to that of most atmospheric particles"?

Done. The reference Ebert et al., 2002 was added.

Page 5, Line 25: the authors might want to add that there were no airborne instruments retrieving refractive index (RI) during DISCOVER-AQ (e.g. DASH-SP or PI-Neph during SEAC4RS) for a direct comparison to their HSRL-2 retrieved RI?

Done. The following sentence has now been added: "*It should be noted that the refractive index was not directly measured in situ during DAQ, although there are instruments capable of inferring this (Shingler, 2016).*"

Page 6, Line 4: The connection between the shallow wintertime boundary layer in California and the larger impact of the spherical filter is not straightforward. The authors should explain/expand more.

The shallow wintertime boundary layer in California limited the number of data points obtained for each vertical profile. Given that most non-spherical particles in urban scenes are mostly observed

closer to the surface (with the exception of dust transport cases), the probability that a data point would be affected by high depolarization and therefore screened out was larger than within the dataset obtained in Texas, because there were fewer data points in California. The text has been modified to discuss this point: *"This screening step has a larger impact on the DAQ CA data set than on the DAQ TX data set because of the shallow wintertime boundary layers observed in California \hl{which limited the number of data points obtained for each vertical profile. Given that most non-spherical particles in urban scenes are mostly observed closer to the surface (except for dust transport cases), the probability that a data point would be affected by the high depolarization and therefore screened out was larger for the CA dataset.}"*

Page 6, Line 17 and Figure 2 & 3: Can the red points (i.e. "Mie retr") be changed to green, as they are retrieved outputs? I also suggest to (i) change the red arrows in black, (ii) change red text in black on figure 3, (iii) delete green arrow and text on fig 3 as this is implied by the red text above, (iv) change mi,j into (mRdry, mIdry) and delete i,j, (v) change Qdry and Qamb in green instead of black, (vi) add "Houston, TX" after "Channel View".
      Done. All suggested changes were made per the reviewer.

Page 6, Line 17: 'good agreement" needs more quantification e.g. within how many percent?
      Done. The sentence now reads: "(…) show good agreement with the measured scattering (within 20%) and absorption (within 2 $Mm^{-1}$) coefficients."

Page 6, Line 24 and 28: "internally mixed" is repeated twice.
      Done. The repetition was removed. The portion from Line 28 now reads: "The hydrated particle refractive index (…)."

Page 7, Line 25: "Examples of profile-to-profile"
      Done.

Page 7, Line 28: is it "10-15%" on all parameters?
      That is correct. We modified the sentence to make that clear: *"(…) but are estimated to be approximately 10-15\% \hl{for all parameters}"*

Page 7, Line 30: "108 coincident profiles (see Section 5.1)" instead of "points"
      Done.

Page 8, Line 1: The reader would benefit from a table showing statistics, equivalent to Table 1 or at least add the correlation coefficients on Figure 6. Comparisons regarding number concentration and effective radius also need to be described in the text.
      Correlation coefficients have been added to Fig 6. We also added text for the comparisons of number concentration and effective radius: *"Correlation coefficients were 0.53 and -0.05 for effective radius, and 0.24 and 0.23 for number concentration respectively for California and Texas."*

Page 8, Line 10: "are expected to be similar" and "<60% RH"
      Done.

Page 8, Line 19: "measured" is repeated twice
     Done.

Page 8, Line 28: please add "vertical integration of ambient extinction coefficient"
     Done.

Page 8, line 26 to Page 9, line 18: the authors use 6 figures (i.e. figure 8, S1, 9, S2, S3, 11) to explain that the coarse mode (as well as the presence of aerosols below the aircraft in California) is the cause of the difference between HSRL-2 and in situ scattering and extinction coefficients. The reasoning could be clarified and some figures could be merged together. For example, why not show the HSRL AOD together with AERONET and in situ AOD on Figure 9? The switching back and forth between supplementary and main figures is not obvious as well. Also Figure 10 should be described in much more detail and paired with the description of figure 8.
     Figures S1 and 9 were merged to produce Fig 10 per the reviewer's suggestion. The description of figure 10 now offers more details. Figure 10 is now figure 9 so its description could flow more naturally following the description of figure 8.

Figure 9, legend: add "vertical integration"
     Done. Figure 9 (now figure 10) caption was modified per the reviewer: "In situ values were obtained by vertical integration of the ambient extinction profile."

Figure 11, legend: "scaled to a maximum aerosol layer height of 1 km for CA.." needs more explanation and (B) should also read AERONET/DRAGON
     Done. More explanation about the scaling of the size distributions were added in the text:
     "AERONET retrievals of size distributions are reported per unit area. The volume distributions from Figure 11A have been converted to represent per unit volume quantities by assuming a maximum aerosol layer height of 1 km and 3 km for California and Texas, respectively. Those values were estimated from the extinction profiles obtained with HSRL-2 during DAQ (see Figure S3)."
     Done. "AERONET/DRAGON" was added to the description of Figure 11B.

Figure S2, legend: should read "ratio of in situ AOD to AERONET AOD"
     Done.

Page 9, Line 23: The authors need to be cautious: "excellent agreement" is a strong statement and does not apply to the HSRL-derived effective radius, for example.
     We have removed effective radius from the statement.

Page 9, Line 25: "within roughly 50% of the insitu values". This does not seem to have been described in the text. Figure 6 needs to show +-50% lines.
     On Page 8 (lines 1-2) we mentioned that the median biases for number concentration were less than 50% for both California and Texas. We have added ± 50% lines to Figure 6 per the reviewer.

**Anonymous Referee #3**

*P8 Line 22-25: The authors provide an explanation for the discrepancy between the HSRL-2 and the calculated in situ, addressing the presents of course mode aerosols. The reviewer agrees this could be a possible explanation and the authors supports their explanation nicely using the AERONET data. However, it is not clear what type of aerosols do the authors suggest could explain this discrepancy, given that depolarization ratio at 532 nm wavelength greater than 5% was screened (meaning the authors have removed dust particles). Also why Texas data has accentuated deviation compared to California? What should be the size of these course mode aerosols? If incorporating the 5 um LAS data was not sufficient to explain the discrepancy. Could there be any other explanation for this discrepancy?

Marine aerosols are usually large and spherical when hydrated and the Houston area is very close to the Gulf of Mexico and the DAQ measurements were obtained during summer when the relative humidity is much higher. So it is plausible that marine aerosols could have caused the discrepancy observed between in situ and lidar measurements. It is also possible, based on partial data, that the aerosols in Texas displayed a complex hygroscopic behavior that could not be properly parameterized by the gamma power-law used to calculate the in situ ambient extinction.

The text has been modified to reflect this explanation: *"Marine aerosols are usually large and spherical particles when hydrated. It is plausible to assume that marine aerosols might have contributed to the discrepancy observed between in situ and lidar measurements due to the close proximity of the Houston area to the Gulf of Mexico and the fact that the DAQ measurements were obtained during summer when the relative humidity is much higher."*

*P6, Line 18 "We assume that the aerosol RI is wavelength independent (450-700nm), this assumption requires a short justification, especially in the lower wavelength range close to the UV. It seems like a reasonable assumption for mostly scattering aerosols (as shown in this study), but needs to be mentioned. Were there any absorbing aerosols present in one of the sites? (e.g. BB aerosols)

We now note that in situ measurements indicate the aerosol to be predominantly scattering with only limited observations of absorbing particles: *"We assume that the aerosol refractive index is wavelength-independent over the spectral range covered by the in situ instruments (i.e. 450 nm to 700 nm) which is a reasonable assumption for aerosols that are mostly of the scattering type. In situ measurements during DAQ CA and TX indicate the aerosols to be predominantly scattering, with limited observations of absorbing particles. The wavelength-independent assumption for the refractive index is also consistent with the lidar retrieval methodology."*

*Was there any chemical information of the aerosols on the airplane platform or ground base that could support the conclusions?

Unfortunately the aircraft composition measurements are limited to water-soluble species (measured with a PILS), which is why they are not discussed in this paper.

*The authors report that the HSRL-2 vertical profile are products within 30 min matched to the spiral data, was the aerosol population well mixed, for this comparison to be meaningful?

According to Anderson et al., (2003), it is reasonable to assume that the aerosol is well mixed at the scales we have chosen as our coincidence criteria. We added this reference in the text when we describe our coincidence criteria.

*P6, line 25: The entire size distribution shifts toward larger diameters", it seems like reasonable assumption, depending on the type of the aerosol. What is the error associated with this assumption?

Without detailed aerosol mixing state information, we are unable to assess the error associated with this assumption.

Specific comments
*Suggest to make figure 6 more clear by moving the headline of DAQ-CA an the DAQ-TX outside (same comment for figure 8)

We have added the headlines to each box to address the reviewer's concern. This avoids compressing the width of what are already small sub-panels.

*P8 Line 7: Please delete brackets from Brock et al., 2016

Done.

*Figure 7: suggest to modify Reff to R_eff (subscript the eff)

Done.

*P9 Line 14: "LAS (0.09-5um diameter) instead of the UHSAS (0.09-5um diameter)". There is a typo: the same range is reported for both instruments.

Done. The range has been corrected.